

# Genetic analysis of water loss of excised leaves associated with drought tolerance in wheat

Ilona Mieczysława Czyczyło-Mysza[1], Izabela Marcińska[1], Edyta Skrzypek[1], Jan Bocianowski[2], Kinga Dziurka[1], Dragana Rančić[3], Radenko Radošević[3], Sofija Pekić-Quarrie[3], Dejan Dodig[4] and Stephen Alexander Quarrie[5,6]

[1] Department of Biotechnology, Polish Academy of Sciences, The Franciszek Górski Institute of Plant Physiology, Kraków, Poland
[2] Department of Mathematical and Statistical Methods, Poznań University of Life Sciences, Poznań, Poland
[3] Faculty of Agriculture, University of Belgrade, Belgrade, Serbia
[4] Maize Research Institute Zemun Polje, Belgrade, Serbia
[5] Newcastle University Business School, Newcastle upon Tyne, United Kingdom
[6] Faculty of Biology, Belgrade University, Belgrade, Serbia

Corresponding author
Ilona Mieczysława Czyczyło-Mysza,
czyczylo-mysza@wp.pl

## ABSTRACT

**Background**. Wheat is widely affected by drought. Low excised-leaf water loss (ELWL) has frequently been associated with improved grain yield under drought. This study dissected the genetic control of ELWL in wheat, associated physiological, morphological and anatomical leaf traits, and compared these with yield QTLs.

**Methods**. Ninety-four hexaploid wheat (*Triticum aestivum* L.) doubled haploids, mapped with over 700 markers, were tested for three years for ELWL from detached leaf 4 of glasshouse-grown plants. In one experiment, stomata per unit area and leaf thickness parameters from leaf cross-sections were measured. QTLs were identified using QTLCartographer.

**Results**. ELWL was significantly negatively correlated with leaf length, width, area and thickness. Major QTLs for ELWL during 0–3 h and 3–6 h were coincident across trials on 3A, 3B, 4B, 5B, 5D, 6B, 7A, 7B, 7D and frequently coincident (inversely) with leaf size QTLs. Yield in other trials was sometimes associated with ELWL and leaf size phenotypically and genotypically, but more frequently under non-droughted than droughted conditions. QTL coincidence showed only ELWL to be associated with drought/control yield ratio.

**Discussion**. Our results demonstrated that measures of ELWL and leaf size were equally effective predictors of yield, and both were more useful for selecting under favourable than stressed conditions.

## INTRODUCTION

Wheat (*Triticum aestivum* L.) is grown worldwide under diverse climatic conditions. Drought is a significant factor restricting wheat production, affecting large areas in both developing and developed countries. Climate change is predicted to reduce rainfall during

wheat reproductive development and grain filling in many countries, so a frequent target for wheat breeders is improving drought tolerance. The influence of water availability on plant productivity suggests that water limitation has shaped the natural variation and evolution of many physiological traits (*Dudley, 1996*).

Many traits have been considered for drought tolerance screening (e.g., *Foulkes et al., 2007*; *Reynolds, Saint Pierre & Vargas, 2007*; *Monneveux, Jing & Misra, 2012*). A physiological trait often used as a selection criterion for improving drought tolerance is rate of leaf water loss. This has been studied in wheat since 1937 (*Bayles, Taylor & Bartel, 1937*), and in excised wheat leaves since 1969 (*Salim, Todd & Stutte, 1969*) as a technique for measuring drought avoidance in cereal seedlings. Since then, excised-leaf water loss (ELWL) in wheat has been used extensively as a selection tool indicating drought tolerance (e.g., *Dedio, 1975*; *Kirkham et al., 1980*; *Clarke & McCaig, 1982b*; *Winter, Music & Porter, 1988*; *Yang, Jana & Clarke, 1991*; *Dhanda & Sethi, 1998*; *Golestani Araghi & Assad, 1998*; *Chandra & Islam, 2003*; *David, 2010*). Thus, rate of water loss from excised leaves has been negatively associated with grain yield under drought in wheat in many reports (e.g., *Clarke & McCaig, 1982a*; *Clarke & Townley-Smith, 1986*; *Clarke et al., 1989*; *Haley, Quick & Morgan, 1993*; *Chandra & Islam, 2003*). Excised-leaf water loss was found to be heritable and predominantly controlled by additive gene effects (*Clarke & Townley-Smith, 1986*; *Dhanda & Sethi, 1998*; *Chandra & Islam, 2003*).

The availability of molecular marker technologies provides opportunities to dissect the genetic control of physiological traits, and gives breeders access to quantitative trait loci (QTLs) for traits suitable for introgression to improve varieties. A detailed analysis of the genetic control of ELWL in bread wheat has not yet been fully reported, though in preliminary results, Yang's group (*Yang et al., 2009*; *Mei, 2012*) identified QTLs for rate of excised-leaf water loss in two wheat recombinant inbred line populations.

Thus, the primary aims of this study were to dissect the genetic control of excised-leaf water loss using a well-characterised wheat mapping population of doubled haploid lines from the cross Chinese Spring × SQ1 (*Quarrie et al., 2005*; *Quarrie et al., 2006*; *Habash et al., 2007*; *Czyczyło-Mysza et al., 2013*), and to identify leaf parameters likely to determine genetic variation in ELWL.

Yield has been recorded in this mapping population in over 50 treatment × site × year occasions including 12 trials where drought treatments caused significant reductions of yield. Additionally, therefore, phenotypic and genetic associations of yield with both ELWL and several leaf parameters were also compared. The value of ELWL as a selection criterion for improving yield in wheat under drought stress is questioned.

## MATERIALS AND METHODS

### Plant material

The mapping population consisted of 94 doubled haploid lines (CSDH) from the cross between hexaploid wheat (*Triticum aestivum* L.) genotypes Chinese Spring (CS) and SQ1 (a breeding line) according to *Quarrie et al. (2005)* and available from the John Innes Centre, Norwich (mike.ambrose@bbsrc.ac.uk).

## Excised leaf dehydration

Excised-leaf water loss in the CSDH population was determined using the procedure of *Clarke & McCaig (1982a)*. After six weeks vernalization at 4 °C, plants were grown under well-watered conditions in a temperature-controlled glasshouse until leaf 4 ligule emerged. This leaf was then detached, quickly transferred to a nearby walk-in growth chamber maintained at 20 °C, 50% relative humidity and continuous light of 250 $\mu$mol m$^{-2}$ s$^{-1}$ (HPS "Agro" lamps, Osram), placed on a v-shaped card support (Fig. S1.) and water loss monitored. Leaf weights were recorded immediately (0 h), after 3 and 6 h and finally after drying at 70 °C for 48 h.

The mapping population was tested three times (Experiment I, II and III) under ostensibly identical conditions in the growth chamber in consecutive years (2007–2009), with three replicate leaves sampled per CSDH line on each occasion.

ELWL after 3 h, 6 h and from 3 to 6 h was calculated as water loss per unit initial water content (ELWLW), according to *Clarke & McCaig (1982a)* as follows

$$ELWLW_{0-3\,h} = \frac{(FW_0 - FW_3)}{(FW_0 - DW)}, \tag{1}$$

$$ELWLW_{3-6\,h} = \frac{(FW_3 - FW_6)}{(FW_3 - DW)}, \tag{2}$$

$$ELWLW_{0-6\,h} = \frac{(FW_0 - FW_6)}{(FW_0 - DW)} \tag{3}$$

where $FW_0$, $FW_3$ and $FW_6$ are fresh weight after 0, 3 and 6 h, respectively, and DW is dry weight after drying at 70 °C.

Leaf length and width were measured before dehydration to estimate leaf surface area (LA) as length $\times$ width $\times$ 0.78: a factor previously determined to be appropriate for wheat leaf four (*Quarrie & Jones, 1977*). From this, rate of water loss/cm$^2$ (ELWLA) during the first 3 h, 3–6 h and 6 h of excision was calculated as for Eqs. (1)–(3), substituting LA for water content, as well as initial leaf FW/cm$^2$.

As ELWL has also been expressed in the literature on the basis of initial leaf FW or DW, ELWL on the basis of FW (ELWLF) and DW (ELWLD) were calculated (substituting FW-DW in Eqs. (1) to (3) with $FW_0$ or DW, respectively).

## Leaf morphological and anatomical measurements

In experiment (III), prior to dehydration the basal *ca.* 2 cm of each leaf 4 was cut and placed into 70% ethanol solution. Leaf cross-sections were prepared manually, and analysed on a microscope slide under a bright-light microscope at $\times 5$ magnification. Sections were photographed with a digital camera (LEICA DC 300) and leaf thickness measured at the midrib and along the lamina: two measurements in the "valley" between two secondary vascular bundles and two measurements at the thickest part across vascular bundles on each side of the midrib. Thus, CSDH line mean midrib thickness and lamina thickness were based on three measurements at the midrib and 24 measurements across the lamina (three leaves $\times$ (four maximum widths+four minimum widths)).

Leaf segments were also examined directly at ×10 magnification to count stomata per field of view (0.761 mm$^2$). Two fields of view either side of the midrib on the lower leaf surface were selected randomly on each leaf segment.

## Grain yield from field and pot trials

Grain yield per plant from field trials with 95 CSDH lines in Norwich, UK in 1997 and 1998 (mean of five random plants per CSDH line) were described in *Quarrie et al. (2005)*, and in Zaječar, Serbia in 2000, 2001 and 2002 (yield and plant number per row) were described in *Quarrie et al. (2005)*, *Quarrie et al. (2006)*. Grain yield per plant from pot trials in Krakow, Poland in 2007, 2008, 2010 and 2011 with irrigated and droughted treatments were described in *Czyczyło-Mysza et al. (2013)* and *Czyczyło-Mysza (2013)*.

Further yield trials were carried out in the field in Zaječar in 2004–2005 and 2005–2006, with plants grown from autumn sowings (30th October, 2004 and 14th November, 2005, respectively) as described for previous years in *Quarrie et al. (2005)*, with two treatments. Two replicate plots were rainfed and two replicate plots were covered with a rain-out shelter at the beginning of tillering, from 4th (2005) and 5th (2006) April, as described in *Dodig et al. (2002)*. The shelter stayed over droughted plots until maturity and reduced light intensity by around 50%.

Pot trials with 94 CSDH lines were carried out in Krakow in 2006, with seedlings transferred to a glasshouse (three plants per line) with or without vernalisation, as described in *Czyczyło-Mysza et al. (2013)*. Only the main ear was sampled for grain weight. Two other trials, under the same conditions described for 2010 and 2011 by *Czyczyło-Mysza et al. (2013)*, were carried out in 2012 and 2013. Yield per plant was recorded in irrigated and droughted treatments with three replicate plants per CSDH line and treatment.

Grain yield per plant was also measured in the following trials. In 1994, all CSDH lines were multiplied in the field from a spring sowing with one row per line at Morley Experimental Station, Norfolk, UK, and in a soil glasshouse trial at the John Innes Centre, Norwich, UK using 73 CSDH lines. Spring-sown plants, one row per CSDH line in two replicate plots, were watered until 16th May and then again from 6th July, to give a drought stress during flowering and early grain filling. Field trials with 95 CSDH lines were sown in the autumn with two replicate plots of three rows per CSDH line in 2002–3 and 2003–4 in Conselice and Idice, northern Italy, organised by Società Italiana Sementi Spa (*Ravaglia, 2005*). The ozone fumigation trial of 2003 using open-top chambers at Newcastle University's Close House Experimental Station, UK (*Quarrie et al., 2006*) included two additional ozone fumigation treatments not reported in *Quarrie et al. (2006)*, of nominally 25 ppb and 50 ppb, with four chambers per treatment and one pot with three plants per CSDH line in each chamber. The Close House ozone fumigation trial was repeated in 2005, with non-filtered air (NFA) and NFA plus 50 ppb ozone treatments. Plants were grown and treated exactly as described in *Quarrie et al. (2006)*, except that ozone fumigation began 8 d earlier, during the rapid tillering phase.

In total, these 52 year × site × treatment trials for yield per plant included 19 regarded as control (little stress) and 12 where a drought treatment was the major stress, reducing

yields significantly, by at least 10%. For these 12 trials a mean drought effect was calculated for each CSDH line as mean drought yield/mean control yield.

A further estimate of yield per plant for each CSDH line under favourable and stressed conditions was calculated using genotype × environment plots for each line as described in *Quarrie et al. (2005)*. Linear regressions of genotype yields from the 52 trials on site mean yields were used to calculate yield per plant for each CSDH line at site mean yields of 2 and 7 g/plant (equivalent to *ca.* 2.5 and 9 t ha$^{-1}$, respectively).

## Phenotypic analysis

Phenotypic data were analysed using GenStat v. 17, and Microsoft EXCEL$^{TM}$ (trait correlations and broad-sense heritabilities). Normality of distributions of CSDH means for each trait were tested using the Shapiro–Wilk normality test (*Shapiro & Wilk, 1965*). Two-way analysis of variance (ANOVA) was carried out for each trait to determine effects of experiments, lines and experiments × lines interaction. Relationships amongst traits were calculated based on CSDH line means.

## QTL analysis

The CS × SQ1 genetic map of *Czyczyło-Mysza et al. (2013)*, with 702 non-duplicated markers (80 RFLP, 227 SSR, 81 AFLP, 292 DArT, 14 EST, five biochemical and three phenotypic markers) distributed on 21 chromosomes, was slightly adjusted by replacing missing marker scores with scores predicted from flanking markers, when these both had the same allele score, and reanalysed to achieve the best-fit marker orders and to remove occasional order inconsistencies in the *Czyczyło-Mysza et al. (2013)* map. This adjusted genetic map of 3,640.5 cM (Kosambi mapping function) was used for QTL analysis. QTLs for CSDH line mean data in each experiment were identified using single-marker analysis (SMA) with Windows QTLCartographer version 2.5 (*Wang, Basten & Zeng, 2011*) or QTL Cartographer v. 1.17j, 28 January 2005 for Macintosh, and Windows QTLCartographer was used for composite interval mapping (CIM) of ELWL and other leaf traits. A QTL from CIM was accepted when the LOD score was greater than 2.

To allow for trait variation from experiment to experiment and to compare QTL coincidences amongst traits on an equal scale, single marker LRmapqtl output was modified by expressing marker additive effects as ratios of the Minimum Significant Additive Effect (MSAE), determined as the minimum absolute one-star [$^\star$] $P \leq 0.05$ additive effect for a particular trait and experiment. Thus, 1 = marker additive effect equal to MSAE. Using this procedure, a marker additive-effect ratio (MAR) of one is equivalent to $P = 0.05$ with LRmapqtl, and $P = 0.01$, 0.001 and 0.0001 with LRmapqtl was determined to be equivalent to ratios of *ca.* 1.32, 1.65 and 1.92, respectively, with these CSDH lines.

For each trait and marker, a mean MAR was calculated as the mean of MARs from each experiment. Thus, the three-experiment mean of MARs (Method 1) was derived as: trait 3-replicate mean → LRmapqtl → marker additive effect → additive-effect ratio → I-III mean MAR (output for ELWLW$_{0-3\,h}$ illustrated in Fig. S2, Method 1). This allowed SMA results for all traits to be compared using the same scale and facilitated graphical display. As peak additive-effect ratios for a particular trait frequently occurred at the same

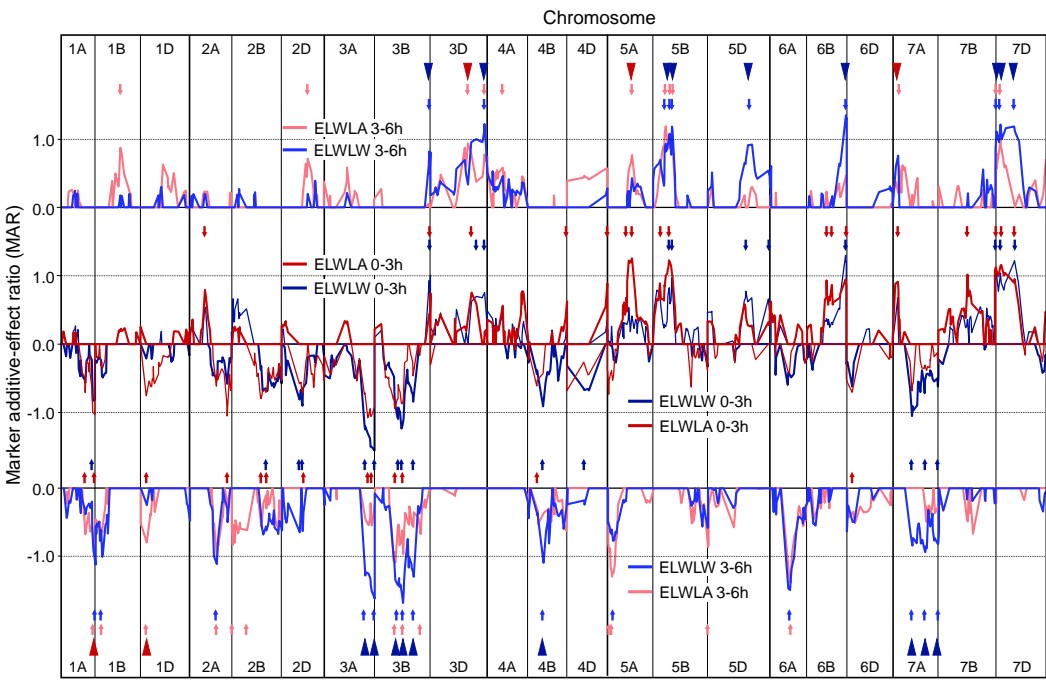

**Figure 1  Single marker analysis (SMA) of additive effects.** SMA for ELWLW 0–3 h , ELWLA 0–3 h, EL-WLW 3–6 h and ELWLA 3–6 h expressed as ratios of the minimum significant additive effect with marker additive ratios (MARs) meaned across the three experiments. Lines join MARs for adjacent markers. The four ELWL traits are grouped according to time period. Continuous coloured lines join MARs for adjacent markers. Markers are ordered sequentially left to right from chromosome 1A short arm to chromosome 7D long arm. Positive MARs indicate alleles with increasing effects from Chinese Spring. Negative MARs indicate alleles with increasing effects from SQ1. Short arrows, coloured according to ELWL trait, identify QTL peaks described in Table S1. Arrowheads indicate coincidence of QTLs for ELWLW 0–3 h and ELWLW 3–6 h (blue), and ELWLA 0–3 h and ELWLA 3–6 h (red).

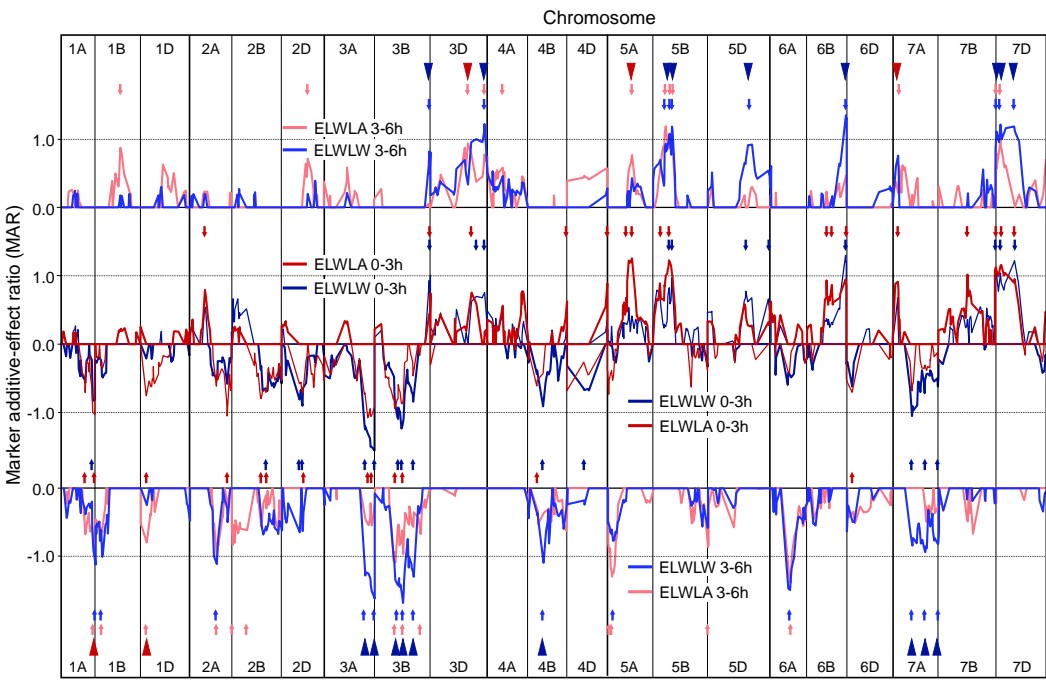

markers in each experiment, though not always reaching one-star (5%) significance with LRmapqtl, the arbitrary threshold of $0.5 \times$ MSAE was used for including a mean MAR for graphical presentation. SMA for traits measured only in Experiment III are also presented as additive-effect ratios. Positive and negative MARs indicate alleles increasing the trait coming from CS (CS alleles increasing) and SQ1 (SQ1 alleles increasing), respectively.

QTLs declared to be present using mean MARs (Method 1) were chosen to be those equivalent to $P \leq 0.05$ significance using marker additive effects from LRmapqtl output for the three-experiment mean ratio (Method 2), determined as follows: trait 3-replicate mean$_I \rightarrow$ trait mean (I+II+III)/3 $\rightarrow$ LRmapqtl $\rightarrow$ marker additive effect $\rightarrow$ MAR (output for ELWLW $_{0-3\ h}$ illustrated in Fig. S2, Method 2). These QTLs were characterised in Table S1, and identified with arrowheads in Figs. 1 and 2, Fig. S2.

A coincidence of QTLs was assumed when SMA MAR maxima and/or CIM LOD score maxima were within 10 cM, representing a minimum precision typical for QTL detection (*Mangin & Goffinet, 1997*).

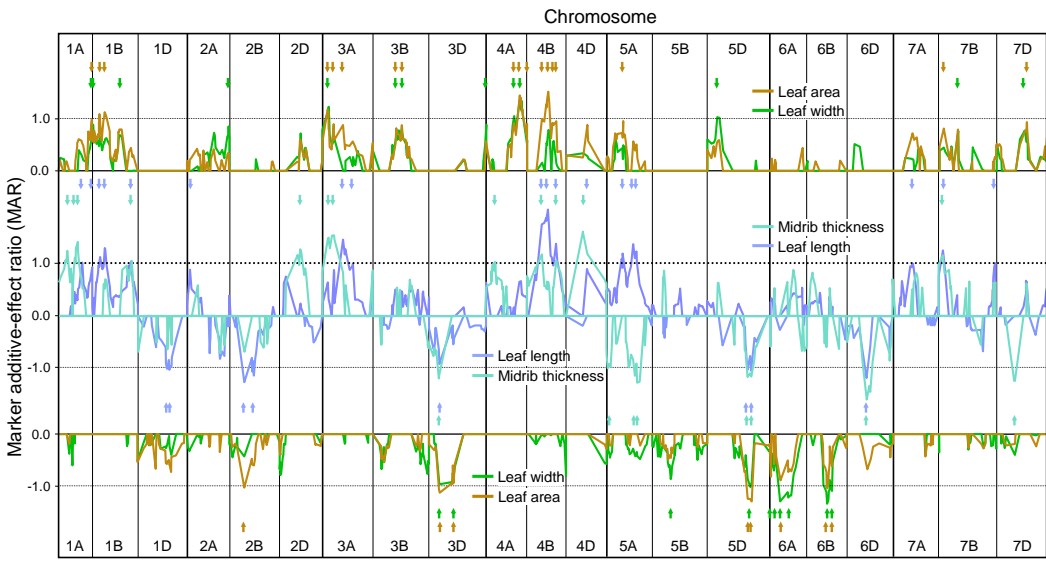

**Figure 2** **Single marker analysis (SMA) of additive effects for leaf length, leaf width, leaf area and leaf midrib thickness.** SMA of additive effects for leaf length, leaf width, leaf area and leaf midrib thickness, expressed as ratios of the minimum significant additive effect with marker additive ratios (MARs) meaned across the three experiments. Leaf traits are grouped according to similarity of their MARs (leaf length and midrib thickness, leaf width and leaf area). Short arrows, coloured according to leaf trait, identify QTL peaks described in Table S1. Other details as described for Fig. 1.

# RESULTS

## Phenotypic analyses
### Phenotypic variation

The Shapiro–Wilk normality test showed all traits to be normally distributed in each year. ANOVA showed the main effects of CSDH lines, experiments and line × experiment interactions to be significant for all traits (Tables 1A, 1B). Multivariate analysis of variance indicated that effects of experiments (Wilk's $\lambda = 0.02257$; $F_{2,14} = 668.25$; $P < 0.0001$), lines (Wilk's $\lambda = 0.00000121$; $F_{93,651} = 8.31$; $P < 0.0001$) and experiments × lines interactions (Wilk's $\lambda = 0.007462$; $F_{186,1302} = 3.00$; $P < 0.0001$) were all highly significant.

Thus, despite growing plants under ostensibly the same conditions and time of year each year, ANOVA showed highly significant genotype × year interactions for every trait (Table 1A). Leaves were generally longer in 2009 and narrower in 2007, leading to leaf areas being 44% greater in 2009 (18.0 cm$^2$) than in 2007 (12.5 cm$^2$), though leaf FW/cm$^2$, a measure of leaf thickness, was only 13% greater in 2009 than in 2007 (17.3 and 19.6 mg/cm$^2$, respectively). The highly significant genotype × year interactions for measures of ELWL were reflected in genotypic correlations for measures of ELWL across the three years, around 40% of which were either non-significant or significant at only $P < 0.05$ (Table S2). Despite the highly significant genotype × year interaction for measures of ELWL, broad-sense heritability for ELWLW$_{0-3}$ $h$ was high each year ($h^2 = $ 0.92, 0.77, 0.63 in 2007, 2008, 2009, respectively). Although highly significant experiment and experiment × line interaction effects were observed, to simplify further description

**Table 1** **(A) Results of two-way analysis of variance for traits measured in experiments I–III. (B) Analysis of variance for traits measured only in Experiment III (2009).**

**(A)**

| Trait | Source of variation | | | | | | | | | | |
|---|---|---|---|---|---|---|---|---|---|---|---|
| | CSDH line | | Year | | CSDH line × Year | | Error | | F-ratio for: | | |
| | d.f. | m.s. | d.f. | m.s. | d.f. | m.s. | d.f. | m.s. | CSDH line | Year | CSDH line × Year |
| 0–3 h water loss | 93 | 0.007914 | 2 | 0.2526 | 186 | 0.002145 | 554 | 0.0008804 | 8.99 | 286.9 | 2.44 |
| 3–6 h water loss | 93 | 0.00244 | 2 | 0.08703 | 186 | 0.0007265 | 555 | 0.0003237 | 7.54 | 268.9 | 2.24 |
| 0–6 h water loss | 93 | 0.01612 | 2 | 0.5511 | 186 | 0.003362 | 554 | 0.001485 | 10.85 | 371.0 | 2.26 |
| Leaf length | 93 | 93.39 | 2 | 1,050.0 | 186 | 11.37 | 556 | 4.495 | 20.77 | 233.7 | 2.53 |
| Leaf width | 93 | 0.05378 | 2 | 2.653 | 186 | 0.01897 | 556 | 0.006906 | 7.79 | 384.1 | 2.75 |
| Leaf area | 93 | 70.51 | 2 | 2,163.7 | 186 | 16.17 | 555 | 6.245 | 11.29 | 346.5 | 2.59 |
| Initial FW (0 h) | 93 | 0.03148 | 2 | 1.463 | 186 | 0.005734 | 554 | 0.002547 | 12.36 | 574.5 | 2.25 |
| Leaf DW (48 h) | 93 | 0.000451 | 2 | 0.03479 | 186 | 0.0001143 | 554 | 0.00005021 | 8.98 | 692.9 | 2.28 |
| Initial leaf FW/cm$^2$ | 93 | 21.283 | 2 | 997.756 | 186 | 8.039 | 554 | 4.170 | 5.10 | 239.25 | 1.93 |
| ELWLW (0–3 h) | 93 | 455.9 | 2 | 7752.5 | 186 | 247.0 | 553 | 59.44 | 7.67 | 130.4 | 4.16 |
| ELWLA (0–3 h) | 93 | 15.32 | 2 | 396.4 | 186 | 8.172 | 555 | 3.032 | 5.05 | 130.8 | 2.70 |
| ELWLW (3–6 h) | 93 | 2280.0 | 2 | 2460.2 | 186 | 602.2 | 547 | 166.6 | 13.69 | 14.8 | 3.61 |
| ELWLA (3–6 h) | 93 | 6.086 | 2 | 81.70 | 186 | 2.606 | 555 | 0.9906 | 6.14 | 82.5 | 2.63 |
| ELWLW (0–6 h) | 93 | 771.1 | 2 | 3157.0 | 186 | 255.8 | 553 | 59.03 | 13.06 | 53.5 | 4.33 |
| ELWLA (0–6 h) | 93 | 30.56 | 2 | 475.8 | 186 | 10.61 | 555 | 3.840 | 7.96 | 123.9 | 2.76 |

All main effects for both factors and for interaction effects for all traits were significant at the $P < 0.001$ level

**(B)**

| Trait | Source of variation | | | | F-ratio | P-value |
|---|---|---|---|---|---|---|
| | CSDH line | | Error | | | |
| | d.f. | m.s. | d.f. | m.s. | | |
| Leaf lamina thickness | 93 | 830.8 | 180 | 204.5 | 4.06 | <0.001 |
| Leaf midrib thickness | 93 | 12,416 | 178 | 6,233 | 1.99 | <0.001 |
| Stomata per field of view | 93 | 43.24 | 180 | 17.34 | 2.49 | <0.001 |

and analysis of results, phenotypic data for the three years were pooled (except for traits measured only in Experiment III).

The parents CS and SQ1 differed significantly for four leaf traits and all measures of ELWL (Table 2), with SQ1 consistently having a greater ELWL than CS. SQ1 also had a significantly smaller initial FW, associated with significantly shorter leaves, leading to smaller leaf areas.

Traits ranged amongst CSDH lines from 1.41-fold for leaf lamina thickness to 3.71-fold for initial FW/cm$^2$ (Table 2). Transgressive segregation amongst DH lines was evident for many traits, as max/min ratios amongst DH lines frequently exceeded parent ratios.

The large variation amongst CSDH lines in leaf size was reflected in ranges amongst lines over 3-fold in initial FW, DW, leaf area and initial leaf FW/cm$^2$. These led to similarly large variation in water loss. However, the variation amongst lines in ELWL was <2.0 for 0–3 h and 0–6 h. Only for the period 3–6 h was the range in ELWL amongst lines at least

**Table 2  Phenotypic variation amongst CSDH lines and their parents Chinese Spring and SQ1 for trait data meaned across Experiments I, II and III.** SD for the parent traits indicates experimental variation. Traits shown in italics were measured only in Experiment III (2009).

| Trait | Parents (mean ± SD) | | | CSDH lines | | | |
|---|---|---|---|---|---|---|---|
| | CS | SQ1 | Ratio SQ1/CS | Mean of 94 | Min | Max | Max/Min |
| 0–3 h water loss (g) | 0.145 (±0.028) | 0.133 (±0.041) | 0.92 | 0.147 | 0.078 | 0.245 | 3.13 |
| 3–6 h water loss (g) | 0.0506 (±0.0170) | 0.0591 (±0.0292) | 1.17 | 0.0604 | 0.0293 | 0.1018 | 3.47 |
| 0–6 h water loss (g) | 0.196 (±0.045) | 0.192 (±0.070) | 0.98 | 0.208 | 0.107 | 0.301 | 2.80 |
| Leaf length (cm) | 34.18 (±1.47) | 24.88 (±4.76)[*] | 0.73 | 29.48 | 20.16 | 40.01 | 1.99 |
| Leaf width (cm) | 0.622 (±0.084) | 0.644 (±0.126) | 1.07 | 0.665 | 0.489 | 0.889 | 1.82 |
| Leaf area (cm$^2$) | 16.70 (±2.99) | 12.63 (±4.04)[*] | 0.76 | 15.47 | 7.68 | 24.07 | 3.13 |
| *Leaf lamina thickness* (μm) | 247.4 (±7.2) | 262.0 (±13.4) | 1.06 | 248.9 | 206.4 | 290.1 | 1.41 |
| *Leaf midrib thickness* (μm) | 668.4 (±62.3) | 589.5 (±14.8) | 0.88 | 672.5 | 524.9 | 805.0 | 1.53 |
| Initial FW (0 h) (g) | 0.331 (±0.067) | 0.237 (±0.092)[*] | 0.72 | 0.299 | 0.131 | 0.459 | 3.50 |
| Leaf DW (g) | 0.042 (±0.014) | 0.038 (±0.017) | 0.90 | 0.039 | 0.018 | 0.058 | 3.18 |
| Initial leaf FW (mg)/cm$^2$ | 19.77 (±0.59) | 18.45 (±2.47) | 1.07 | 19.26 | 12.42 | 46.08 | 3.71 |
| *Stomata per field of view* | 25.00 (±5.02) | 26.33 (±4.80) | 1.05 | 25.45 | 15.25 | 35.50 | 2.33 |
| ELWLW (0–3 h) | 50.22 (±4.94) | 68.35 (±5.93)[*] | 1.36 | 58.07 | 44.54 | 85.09 | 1.91 |
| ELWLA (0–3 h) | 9.16 (±1.17) | 10.95 (±0.73)[*] | 1.20 | 10.10 | 7.53 | 14.84 | 1.97 |
| ELWLW (3–6 h)[a] | 34.84 (±9.20) | 91.57 (±10.02)[*] | 2.63 | 60.03 | 27.80 | 92.40 | 3.32 |
| ELWLA (3–6 h) | 3.11 (±0.70) | 4.66 (±0.98)[*] | 1.50 | 4.04 | 2.42 | 6.05 | 2.50 |
| ELWLW (0–6 h) | 67.29 (±6.90) | 97.02 (±3.42)[*] | 1.44 | 81.17 | 61.73 | 99.12 | 1.61 |
| ELWLA (0–6 h) | 10.40 (±4.37) | 12.96 (±4.52)[**] | 1.25 | 11.86 | 8.76 | 16.09 | 1.84 |

**Notes.**

Significance levels: [*]$P \leq 0.05$, [**]$P \leq 0.01$ indicate significance of differences between parents using a paired-sample $t$-test for traits measured in all three experiments. No parental means were significantly different for traits measured only in Experiment III.

[a]Calculated on the basis of leaf water content after 3 h.

2.5. ELWLW$_{3-6\,h}$ showed the largest ranges amongst both lines (3.32-fold) and parents (2.63). Ranges for ELWLF (FW basis) and ELWLD (DW basis) showed similar trends, with 0–3 h, 3–6 h and 0–6 h ranges of 1.94-, 2.63- and 1.65-fold, respectively, for ELWLF and 2.25-, 2.86- and 2.14-fold, respectively, for ELWLD. Measures of leaf thickness showed relatively little variation amongst the lines (*ca.* 1.5-fold).

*Phenotypic associations amongst leaf traits*

Leaf parameters associated with leaf size were highly significantly positively correlated with each other (Table 3). The only trait not correlated with leaf length was leaf lamina thickness. The structural support provided by the midrib led to midrib thickness being significantly positively associated with leaf length, width, area, and lamina thickness. All these leaf size parameters were also highly significantly positively correlated with leaf initial FW, DW and initial FW/cm$^2$. Stomatal number/unit area was significantly negatively associated with leaf length, and weakly positively associated with lamina thickness.

All measures of ELWL were highly significantly positively correlated with each other for 0–3 h and 0–6 h, and most ELWL for 3–6 h (Table 4). In general, correlations between ELWL$_{0-6\,h}$ and other leaf parameters were very similar to those for ELWL$_{0-3\,h}$, but less significant, so ELWL$_{0-6\,h}$ is not discussed further. For 3–6 h, only ELWLW was highly

**Table 3 Correlation coefficients for associations amongst leaf traits, using data for each CSDH line and trait meaned across the three experiments, except leaf lamina, midrib thickness and stomatal number/unit area (Experiment III), for which Experiment III data only were used for all correlations).**

| | Leaf length | Leaf width | Leaf area | Lamina thickness | Midrib thickness | Initial FW | Leaf DW | Initial FW/cm$^2$ |
|---|---|---|---|---|---|---|---|---|
| Leaf width | **0.326**** | | | | | | | |
| Leaf area | **0.784***** | **0.834***** | | | | | | |
| Leaf lamina thickness | 0.197 | **0.611***** | **0.515***** | | | | | |
| Leaf midrib thickness | **0.372**** | **0.453***** | **0.506***** | **0.588***** | | | | |
| Leaf initial FW$_{(0\,h)}$ | **0.780***** | **0.734***** | **0.929***** | **0.584***** | **0.585***** | | | |
| Leaf DW | **0.697***** | **0.805***** | **0.927***** | **0.602***** | **0.566***** | **0.914***** | | |
| Initial leaf FW/cm$^2$ | **0.781***** | **0.715***** | **0.917***** | **0.574***** | **0.580***** | **0.999***** | **0.891***** | |
| Stomatal number/unit area | −0.352*** | 0.167 | −0.090 | **0.203*** | −0.202 | −0.131 | −0.007 | −0.146 |

**Notes.**
Significant positive correlations are shown in bold, and significant negative correlations are shown underlined.
*, **, ***, **** indicate correlations significant at $P < 0.05$, $P < 0.01$, $P < 0.001$ and $P < 0.0001$, respectively, with 92 df.

**Table 4 Correlation coefficients for associations amongst measures of ELWL and leaf traits.**

| Trait: | ELWL trait: Time period: | ELWLW 0–3 h | ELWLF 0–3 h | ELWLD 0–3 h | ELWLA 0–3 h | ELWLW 3–6 h | ELWLF 3–6 h | ELWLD 3–6 h | ELWLA 3–6 h |
|---|---|---|---|---|---|---|---|---|---|
| ELWLW 0–3 h | | 1 | | | | | | | |
| ELWLF 0–3 h | | **0.991***** | 1 | | | | | | |
| ELWLD 0–3 h | | **0.761***** | **0.820***** | 1 | | | | | |
| ELWLA 0–3 h | | **0.772***** | **0.807***** | **0.852***** | 1 | | | | |
| ELWLW 3–6 h | | **0.836***** | **0.830***** | **0.594***** | **0.658***** | 1 | | | |
| ELWLF 3–6 h | | **0.209*** | **0.206*** | 0.120 | **0.209*** | **0.687***** | 1 | | |
| ELWLD 3–6 h | | 0.138 | 0.169 | **0.298*** | **0.299**** | **0.581***** | **0.925***** | 1 | |
| ELWLA 3–6 h | | 0.174 | 0.195 | **0.231*** | **0.406***** | **0.620***** | **0.923***** | **0.932***** | 1 |
| 0–3 h water loss | | **0.219*** | **0.271**** | **0.446***** | **0.461***** | **0.223*** | 0.105 | **0.225*** | **0.259*** |
| 3–6 h water loss | | −0.059 | −0.029 | 0.078 | 0.169 | **0.346***** | **0.712***** | **0.741***** | **0.773***** |
| Leaf length | | −0.406**** | −0.371*** | −0.139 | −0.212* | −0.340*** | −0.108 | 0.010 | −0.020 |
| Leaf width | | −0.231* | −0.219* | −0.170 | −0.185 | −0.135 | 0.028 | 0.016 | 0.012 |
| Leaf area | | −0.385**** | −0.359*** | −0.199 | −0.250* | −0.279** | −0.036 | 0.023 | −0.001 |
| Leaf lamina thickness | | −0.258* | −0.224* | −0.132 | −0.016 | −0.096 | 0.118 | 0.162 | **0.221*** |
| Leaf midrib thickness | | −0.351*** | −0.326** | −0.187 | −0.084 | −0.248* | −0.013 | 0.077 | 0.123 |
| Leaf initial FW | | −0.381*** | −0.336*** | −0.061 | −0.035 | −0.282** | −0.032 | 0.106 | 0.132 |
| Leaf DW | | −0.376*** | −0.365*** | −0.304** | −0.176 | −0.238* | 0.010 | −0.001 | 0.095 |
| Leaf initial FW/cm$^2$ | | −0.377*** | −0.327** | −0.027 | −0.015 | −0.284** | −0.037 | 0.120 | 0.136 |
| Stomatal number/unit area | | 0.034 | 0.013 | −0.119 | −0.049 | 0.116 | 0.159 | 0.035 | 0.097 |

**Notes.**
Significant positive correlations are shown in bold, and significant negative correlations are shown underlined.
*, **, ***, **** indicate correlations significant at $P < 0.05$, $P < 0.01$, $P < 0.001$ and $P < 0.0001$, respectively, with 92 df.

significantly positively correlated between all measures of ELWL for 0–3 h. Correlations of ELWLF, ELWLD and ELWLA for 3–6 h (except ELWLA$_{3-6\,h}$ and ELWLA$_{0-3\,h}$) with other measures of ELWL for 0–3 h were either non-significant or only weakly significant.

Correlations between measures of ELWL and other leaf parameters showed clear patterns. Thus, $ELWLW_{0-3\,h}$ and $ELWLF_{0-3\,h}$ were both highly significantly ($P < 0.002$) negatively associated with leaf length, area and midrib thickness (larger leaves had lower ELWL) as well as initial leaf FW and DW (Table 4). Correlations between both ELWLD and ELWLA and leaf size parameters were either not significant or only weakly significant (negatively between $ELWLD_{0-3\,h}$ and leaf DW). For ELWLA, significant correlations ($P < 0.05$) occurred with leaf length and area (negative for 0–3 h) and lamina thickness (positive, 3–6 h). Most correlations of $ELWL_{3-6\,h}$ with measures of leaf size were non-significant. Only $ELWLW_{3-6\,h}$ was significantly correlated (negatively) with leaf size parameters: leaf length, area, midrib thickness, initial FW and DW (Table 4). $ELWLF_{3-6\,h}$ and $ELWLD_{3-6\,h}$ were not correlated with any leaf parameters.

### Phenotypic associations of ELWL and other leaf traits with yield

Although yield was not measured in these excised-leaf experiments, yield/plant of the CSDH lines was measured in 52 other year × site × treatment trials. Amongst these, 12 experiments had control (rainfed or irrigated) and at least one effective drought treatment (10–71% yield reduction). Mean yield/plant for these droughted treatments as well as the corresponding mean control yields (augmented with control (non-stressed) treatments from seven other trials) were used to calculate drought-induced yield effect for each CSDH line (expressed as droughted/control yield). Yield/plant for each line at site mean yields of 2 and 7 g/plant was also calculated, as described in Materials and Methods. Associations between these five measures of yield and yield response to drought and measures of ELWL and non-ELWL leaf traits were analysed.

No measure of ELWL was significantly correlated with drought-induced yield reduction (Table 5A). $ELWL_{0-3\,h}$ was more frequently significantly negatively correlated with measures of yield/plant than 0–6 h data (Table S3), and 3–6 h data were the least associated with measures of yield/plant: only $ELWLW_{3-6\,h}$ and $ELWLF_{3-6\,h}$ were significantly correlated with droughted yield/plant and yield/plant at 7 g yield/plant. No ELWLD data were significantly correlated with any measure of yield/plant for any time period.

Overall, measures of ELWL showing the most consistent significant correlations (negatively) with measures of yield/plant were $ELWLW_{0-3\,h}$ and $ELWLA_{0-3\,h}$, with $ELWLA_{0-3\,h}$ being correlated at $P < 0.001$ with yield/plant at 7 g/plant (Table 5A). $ELWLW_{0-3\,h}$ and $ELWLF_{0-3\,h}$ were more significantly negatively correlated with yield/plant under droughted than control conditions. Thus, higher yield under drought was associated with lower ELWLW and ELWLF. However, $ELWLW_{0-3\,h}$ and $ELWLF_{0-3\,h}$ were equally significantly negatively correlated ($P < 0.005$) with yield/plant estimated at site yields of 2 and 7 g/plant.

Correlations of non-ELWL leaf traits with yield/plant were positive. Thus, leaf length and area, initial leaf FW and DW were usually significantly correlated with measures of yield/plant. Leaf length was highly effective at predicting yield/plant under droughted conditions ($P < 0.0001$), and leaf four area was highly effective at predicting yield/plant under favourable conditions (site yield of 7 g/plant), $P < 0.0001$ (Table 5B). Stomatal

**Table 5  Associations of yield/plant and drought-induced yield reduction with measures of ELWL (A) and with leaf four traits (B).** Other details as in Table 3.

**(A)**

| Yield trait:        ELWL trait: Time period: | ELWLW 0–3 h | ELWLF 0–3 h | ELWLD 0–3 h | ELWLA 0–3 h | ELWLW 3–6 h | ELWLF 3–6 h | ELWLD 3–6 h | ELWLA 3–6 h |
|---|---|---|---|---|---|---|---|---|
| Yield/plant - control | −0.216* | −0.195 | −0.173 | −0.304** | −0.103 | 0.055 | 0.058 | −0.017 |
| Yield/plant - droughted | −0.318** | −0.290** | −0.133 | −0.242* | −0.212* | 0.007 | 0.102 | 0.031 |
| Ratio drought/control | −0.123 | −0.120 | 0.020 | 0.048 | −0.127 | −0.050 | 0.044 | 0.047 |
| Yield/plant at 7 g/plant | −0.327** | −0.300** | −0.155 | −0.351*** | −0.233* | 0.000 | 0.074 | −0.062 |
| Yield/plant at 2 g/plant | −0.306** | −0.283** | −0.184 | −0.317** | −0.145 | 0.122 | 0.167 | 0.063 |

**(B)**

| Yield trait:        Leaf trait: | Leaf length | Leaf width | Leaf area | Lamina thickness | Midrib thickness | Initial FW | Leaf DW | Stomatal number |
|---|---|---|---|---|---|---|---|---|
| Yield/plant - control | 0.304** | 0.309** | 0.373*** | 0.211* | 0.108 | 0.260* | 0.288** | −0.079 |
| Yield/plant - droughted | 0.399**** | 0.142 | 0.326** | 0.183 | 0.175 | 0.315** | 0.248* | −0.231* |
| Ratio drought/control | 0.043 | −0.185 | −0.084 | −0.027 | 0.058 | 0.024 | −0.073 | −0.149 |
| Yield/plant at 7 g/plant | 0.383*** | 0.300** | 0.415**** | 0.158 | 0.182 | 0.324** | 0.280** | −0.204* |
| Yield/plant at 2 g/plant | 0.250* | 0.273** | 0.318** | 0.247* | 0.144 | 0.242* | 0.230* | −0.088 |

**Notes.**

Significant positive correlations are shown in bold, and significant negative correlations are shown underlined.

*, **, ***, **** indicate correlations significant at $P < 0.05$, $P < 0.01$, $P < 0.001$ and $P < 0.0001$, respectively, with 92 df.

number/mm$^2$ was weakly negatively correlated ($P < 0.05$) with droughted yield/plant and yield/plant under favourable conditions (site yield of 7 g/plant).

### Leaf 4 ELWL and length as selection criteria to improve yield

As ELWL has frequently been suggested to be a useful trait for selecting improved yield under drought, $ELWLW_{0-3\,h}$ was compared with leaf length for their effectiveness at identifying higher-yielding CSDH lines. For each of the two traits, the 94 lines were ranked according to either increasing ELWL or decreasing leaf length and, for each water loss experiment, yields/plant for the 10 lines with the lowest $ELWLW_{0-3\,h}$ and 10 lines with the longest leaves were compared with yields/plant for both the 10 lines at the opposite end of the rankings (highest $ELWLW_{0-3\,h}$ and shortest leaves) and the remaining 84 CSDH lines. Yields were compared for five groups: mean yields from all 52 trials, control mean yields (19 trials), droughted mean yields (12 trials) and yields at site mean yields of 7 g/plant and 2 g/plant (Table 6).

Selecting the 10 most favourable and 10 least favourable CSDH lines resulted in highly significant yield differences for all 52 trials and the 19 control trials with both $ELWLW_{0-3\,h}$ and leaf length in each of the three experiments.

Comparing the 10 most favourable lines with the remaining 84 lines, yield advantages of selecting CSDH lines with the lowest $ELWLW_{0-3\,h}$ were overall small for each yield group, averaging 3.1% across the five yield groups for $ELWLW_{0-3\,h}$, and significant only when meaned across the three experiments, for 52 trial mean yields, as well as the 19 control trials and a site mean yield of 2 g/plant (Table 6). Selecting for $ELWLW_{0-3\,h}$ gave no yield advantage with the 12 droughted trials.
**Table 6 Ratios between yield/plant for the 10 selected CSDH lines and other CSDH lines.** Ratios between yield/plant for the 10 CSDH lines with the lowest ELWLW 0–3 h and with the highest leaf length and yield/plant for both the 10 CSDH lines with the highest ELWLW 0–3 h and with the lowest leaf length and the remaining 84 CSDH lines for five measures of yield/plant.

| Trait | Year | Mean of all 52 trials | Mean of 19 control trials | Mean of 12 droughted trials | Site yield of 7 g/plant | Site yield of 2 g/plant |
|---|---|---|---|---|---|---|
| ELWLW 0–3 h 10 lowest versus 10 highest lines | 2007 | *1.172*\*\*\*\* | *1.153*\*\*\*\* | *1.221*\* | *1.197*\* | 1.114 |
| | 2008 | *1.086*\*\*\*\* | *1.103*\*\*\* | 1.052 | 1.113 | 1.083 |
| | 2009 | *1.140*\*\*\*\* | *1.103*\*\*\*\* | *1.233*\*\* | 1.109 | *1.141*\* |
| | mean | *1.200*\*\*\*\* | *1.156*\*\*\*\* | *1.252*\*\* | *1.240*\*\* | *1.151*\*\* |
| Leaf length 10 highest versus 10 lowest lines | 2007 | *1.106*\* | *1.114*\*\* | 1.219 | 1.046 | 1.078 |
| | 2008 | *1.250*\*\*\*\* | *1.294*\*\*\*\* | 1.335 | *1.256*\*\* | *1.181*\*\* |
| | 2009 | *1.193*\*\*\*\* | *1.202*\*\*\*\* | 1.247 | 1.142 | *1.163*\* |
| | mean | *1.209*\*\*\*\* | *1.238*\*\*\*\* | 1.276 | *1.202*\* | *1.170*\*\* |
| ELWLW 0–3 h 10 lowest versus 84 remaining lines | 2007 | 1.065 | 1.048 | 1.083 | 1.086 | 1.041 |
| | 2008 | 1.004 | 1.029 | 0.977 | 1.025 | 1.017 |
| | 2009 | 1.024 | 1.011 | 1.030 | 0.984 | 1.036 |
| | mean | *1.0310*\*\*\* | *1.0290*\*\*\*\* | 1.0300 | 1.0317 | *1.0313*\*\* |
| Leaf length 10 highest versus 84 remaining lines | 2007 | 0.978 | 0.943 | 1.026 | *0.893*\*\* | 0.993 |
| | 2008 | 1.028 | 1.024 | 1.049 | 0.976 | 1.016 |
| | 2009 | 1.054 | 1.028 | 1.077 | 0.983 | 1.040 |
| | mean | 1.0200 | 0.9983 | *1.0507*\*\* | 0.9507 | 1.0163 |

**Notes.**

Significant differences[§] are indicated by ratios in bold italics, and ratios less than one are shown in red.

[§]Significance of differences in yield/plant between the 10 most favourable and 10 least favourable CSDH lines for the 52 trials, 19 control and 12 droughted trials were tested using a paired-sample $t$-test with trial means. Significance of differences in yield/plant between the most favourable 10 CSDH lines and the remaining 84 lines for the 52 trials, 19 control and 12 droughted trials were tested using two-way ANOVA, with experiments as replications. Significance of differences in yield/plant between the 10 most favourable and both the 10 least favourable lines and the remaining 84 lines at site yields of 7 and 2 g/plant were tested using a two-sample $t$-test with equal variances.

\*, \*\*, \*\*\*, \*\*\*\*Means of yield/plant for 10 CSDH most favourable lines and either the 10 least favourable lines or the remaining 84 lines significantly different at $P < 0.05, 0.01, 0.001, 0.0001$, respectively.

Leaf length was, overall, a less effective selection criterion for increasing yield/plant (0.7% over the five yield measures), and in 2007, selecting for longer leaves resulted in a significant reduction in mean yield/plant at a site mean yield of 7 g/plant. Nevertheless, leaf length was more effective at increasing yields in the droughted group ($P < 0.05$), giving a mean yield advantage under droughted conditions of 5.1% (Table 6). Selecting for large leaf area similarly gave small and inconsistent yield benefits, averaging 2.3% higher yields over the five yield measures, compared with the 84 other lines (Table S4).

## Genetic analyses

To reduce the complexity of genetic analysis of ELWL and leaf traits, only 3-experiment mean data for the four ELWL traits $ELWLW_{0-3 h}$, $ELWLA_{0-3 h}$, $ELWLW_{3-6 h}$ and $ELWLA_{3-6 h}$, as well as 3-experiment mean data for leaf length, width, area and Experiment III data for midrib thickness are described in detail here. CIM was used with 3-expt-mean data, and QTL peaks coincident between SMA and CIM are identified in Table S1. About 20% of QTLs classified as significant by SMA were also significant using CIM.

### Genetic analysis of ELWL

Detailed genetic analysis using SMA focused on ELWLW and ELWLA for both 0–3 h and 3–6 h, using ratio means as described for Method 1 (Materials and Methods). Figure S2 demonstrates that SMA using Methods 1 and 2 gave very similar QTLs. $ELWL_{0-3 h}$ peak maxima at 25 markers with the most significant MARs for $ELWL_{0-3 h}$ 3-experiment-mean phenotypic data using Method 2 were highly significantly correlated ($r_{23df} = 0.997$) with 3-experiment-mean MARs at the same markers using Method 1. From this regression, a ratio of 1.0 for significance at $P = 0.05$ using Method 2 was equivalent to *ca.* 0.7 using Method 1.

Numbers of QTLs identified for ELWL traits using SMA varied from 20 to 30 (Table S1 and Fig. 1). For $ELWLW_{0-3 h}$, 25 QTLs were identified on 13 chromosomes, with major effects distal on 3AL (SQ1 high ELWLW allele) and 6BL, as well as 7DS (both CS high water loss alleles). Thirty QTLs were identified for $ELWLA_{0-3 h}$, distributed on 17 chromosomes, of which two major QTLs were located on 5A at the *vrn-A1* locus and 5BL, both with CS alleles increasing ELWLA. $ELWLW_{3-6 h}$ gave 24 QTL distributed on 14 chromosomes, with CS alleles increasing ELWLA at major QTLs on 3DL, 5BL, 6BL and 7DS, and SQ1 contributing increasing alleles at major QTLs on 3AL (two QTLs), 3B (three QTLs) and 6AL. $ELWLA_{3-6 h}$ gave 25 QTLs located on 14 chromosomes. SQ1 alleles increased $ELWLA_{3-6 h}$ at two major QTLs, on chromosomes 5AS and 6AL.

The four ELWL traits showed considerable similarities in MAR line traces and locations of QTLs, with 11 QTLs coincident between ELWLW and ELWLA for a particular time period (coincident arrowheads in Fig. 1) and 23 QTLs coincident between time periods (triangles in Fig. 1). Ratios were significant and traces very similar distal on 3AL, 3B, distal on 3DL, 4B, 5B, 6A, distal on 6BL and 7D.

Few QTLs were stably expressed each year, giving MARs $\geq 1$ ($P \leq 0.05$) in all experiments (Table S1, illustrated for $ELWLW_{0-3 h}$ in Fig. S5): $ELWLW_{0-3h}$ - 3, $ELWLA_{0-3 h}$ - 2, $ELWLW_{3-6 h}$ - 8, $ELWLA_{3-6 h}$ - 1. Nevertheless, within experiments, genetic control of ELWL was consistent across a range of time intervals. Thus, in 2007, leaf weights were also recorded after water loss for 8 h. Fourth-order polynomials were fitted to leaf weights sampled at 0, 3, 6 and 8 h to calculate water loss after 1 h, and ELWLW calculated for the intervals 0–1 h, 1–3 h, 3–6 h and 6–8 h. Eighteen genomic regions showed coincidence for all four time intervals (boxed in Fig. S6).

Comparing $ELWLF_{0-3 h}$ and $ELWLD_{0-3 h}$ with $ELWLW_{0-3 h}$ (Fig. S3) showed QTLs with peak ratios $\geq 1.0$ (22 for $ELWLF_{0-3 h}$ and 23 for $ELWLD_{0-3 h}$) largely coincident with those for $ELWLW_{0-3 h}$.

### Genetic analysis of constitutive leaf traits

SMA of leaf length, width, area and midrib thickness demonstrated MAR similarities between leaf length and midrib thickness, and between leaf width and area (Fig. 2). Numbers of QTLs identified using SMA for leaf constitutive traits were similar to those for ELWL: length—26, width—23, area—26, thickness—20 (Table S1). Leaf length QTLs were identified on 14 chromosomes, with the majority of QTLs having increasing alleles from CS. By far the largest QTL for leaf length (CS alleles increasing) was on 4B, very close to the

dwarfing gene *Rht-B1*. Fifteen chromosomes had QTLs for leaf width, though the majority of QTLs were weak (MARs <1.0). Major leaf width QTLs were present on chromosomes 3A, 4A, 6A and 6B, alleles increasing leaf width from CS on 3A and 4A, and from SQ1 on 6A and 6B. Leaf area QTLs were distributed amongst 14 chromosomes, with major additive effects on 4A, 4B (CS alleles increasing) and 5D (SQ1 alleles increasing). MARs more closely followed those for leaf width than for leaf length (Fig. 2). Twenty QTLs for midrib thickness were distributed amongst 13 chromosomes, with major QTLs located on chromosomes 1A, 2D, 3A, 4D (CS alleles increasing) and 5A and 6D (SQ1 alleles increasing).

### Coincidence of QTLs between ELWL and constitutive leaf traits

As phenotypic correlations between ELWL and constitutive leaf traits were almost invariably negative (Table 4), the coincidence of QTLs for ELWL and constitutive leaf traits is compared (Figs. 3A and 3B) with traces of MARs inverted for $ELWLW_{0-3\,h}$, $ELWLA_{0-3\,h}$, $ELWLW_{3-6\,h}$ and $ELWLA_{3-6\,h}$. Although phenotypic correlations between leaf constitutive traits and measures of ELWL were much more significant for ELWLW than for ELWLA, QTLs were coincident between all measures of ELWL and each of the four constitutive leaf traits: QTL coincidences with $ELWLW_{0-3\,h}$ were leaf length—4, width— 2, area—5, thickness—4; $ELWLA_{0-3\,h}$ with length—2, width—4, area—4, thickness—3; $ELWLW_{3-6\,h}$ with length—6, width—3, area—6, thickness—3, and $ELWLA_{3-6\,h}$ with length—2, width—2, area—4, thickness—1. All four measures of ELWL were coincident with leaf trait QTLs distal on 1AL and near 3B centromere (Fig. 3A). QTLs specific for only ELWLW were coincident with leaf trait QTLs on 4B at the dwarfing gene *Rht-B1*, 5DL, 7AL and 7DS.

### Genetic analysis of yield per plant

Figures 4A and 4B shows MARs from SMA for five measures of yield (control, droughted, droughted/control, yield at 7 and 2 t ha$^{-1}$), together with mean MARs for all 52 yield trials, $ELWLW_{0-3\,h}$ and $ELWLA_{0-3\,h}$. Yield QTLs combined using Method 1 from all trials were consistently present with increasing alleles from CS on chromosomes 1D, 4A, 4B, 4D, 5A, 7A, 7B, and from SQ1 on chromosomes 1D, 2B, 2D, 3D, 4B, 5D, 6B and 7A.

Many peak MARs were consistent across the four measures of yield/plant (arrowheads in Fig. 4) but differed from those for yield drought/control ratio (Fig. 4). Major differences in QTLs between the 19 control and 12 droughted yields were present on 16 chromosomes. Increasing alleles were contributed by CS for control-specific QTLs on chromosomes 4D, 5D and 7B, and by SQ1 on chromosomes 1A, 2B, 3B, 3D, 4B and 7A. Drought-specific QTLs were present on 1A, 1D, 4B and 5A (increasing allele from CS), as well as 1D, 2A, 3D and 6B (SQ1 alleles increasing). QTLs with MARs ≥1 for yield at a site yield of 7 t ha$^{-1}$, were located on chromosomes 1A (CS alleles increasing) and 2B, 3D, 4D and 7A (SQ1 alleles increasing). For yield at a site yield of 2 t ha$^{-1}$, QTLs were found on chromosomes 1B, 2B, 4A and 7A (CS alleles increasing), as well as 2D, 3B and 6A (SQ1 alleles increasing).

Yield response to drought (drought/control) showed highly significant ($P < 0.001$) QTLs on chromosomes 1D and 5D (increasing alleles from CS and SQ1, respectively), with other major QTLs ($P < 0.01$) on chromosomes 1A, 3A, 4B and 7A (increasing alleles from CS), and 2A and 6B (increasing alleles from SQ1).
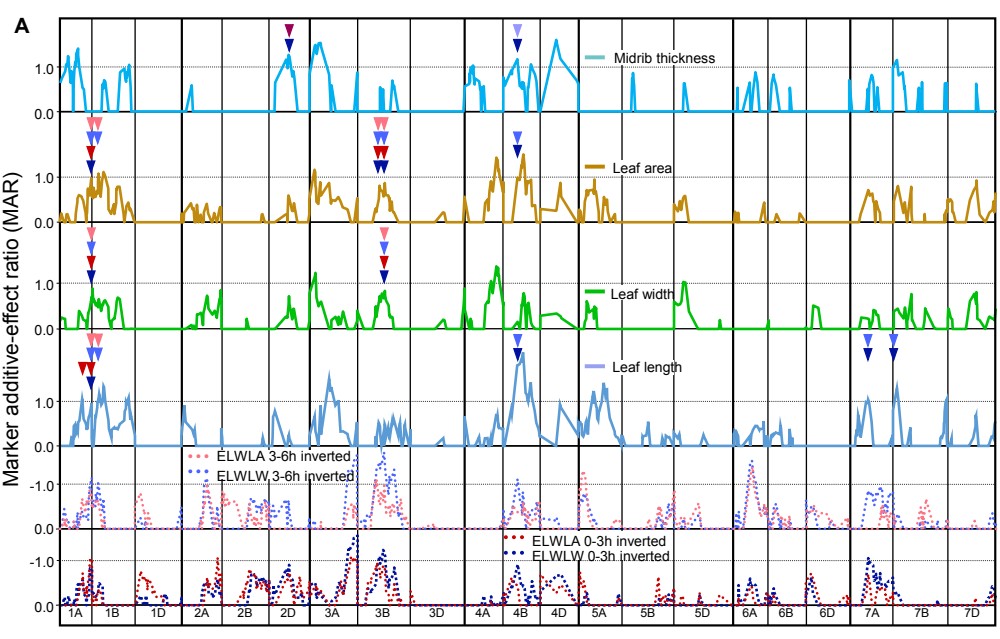

Chromosome

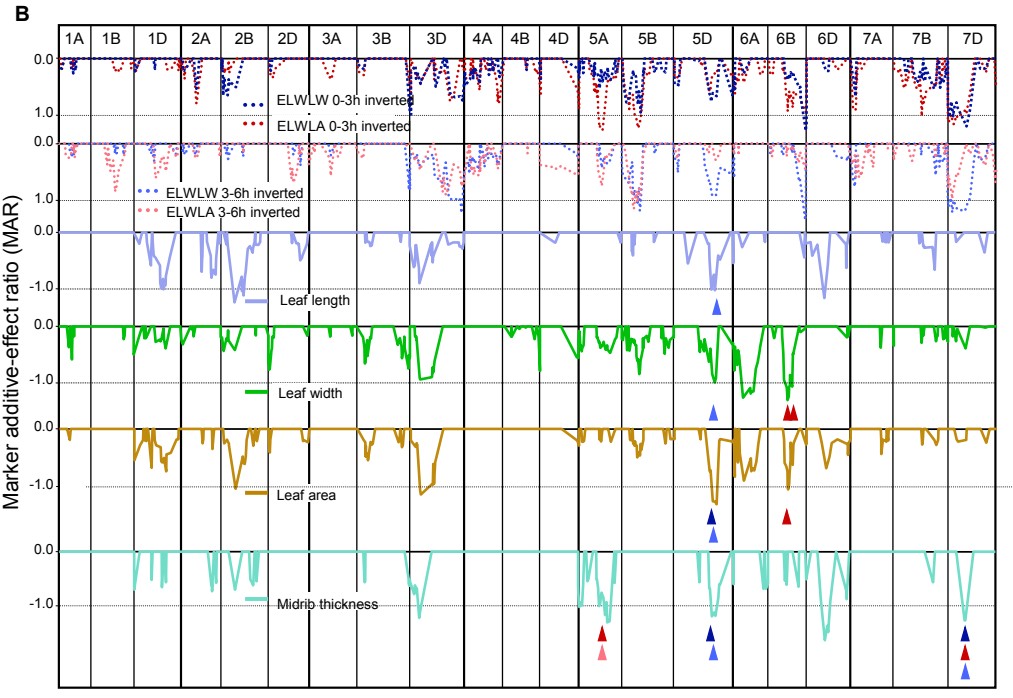

**Figure 3** **Single marker analysis (SMA) of additive effects for both ELWL and leaf traits combined.**
(A) positive marker additive ratios (MARs) with increasing alleles from Chinese Spring, and (B) negative MARs with increasing alleles from SQ1. Note, because of the negative correlations between ELWL and leaf traits (Table 4), ELWL traces are inverted to ease comparison amongst traits. Arrowheads, coloured according to ELWL trait, indicate coincidence between ELWL and leaf trait QTLs. Other details as described for Fig. 1.

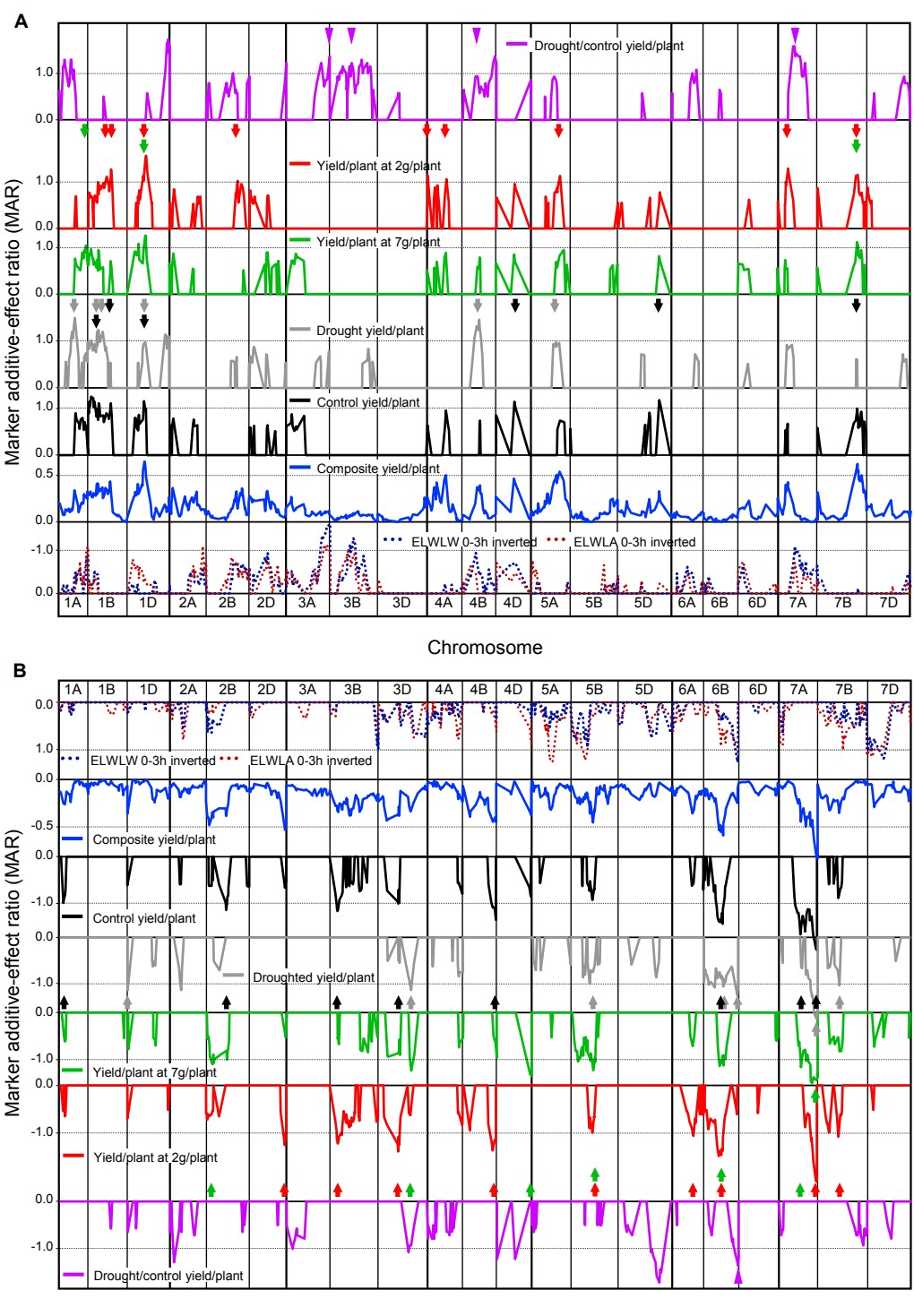

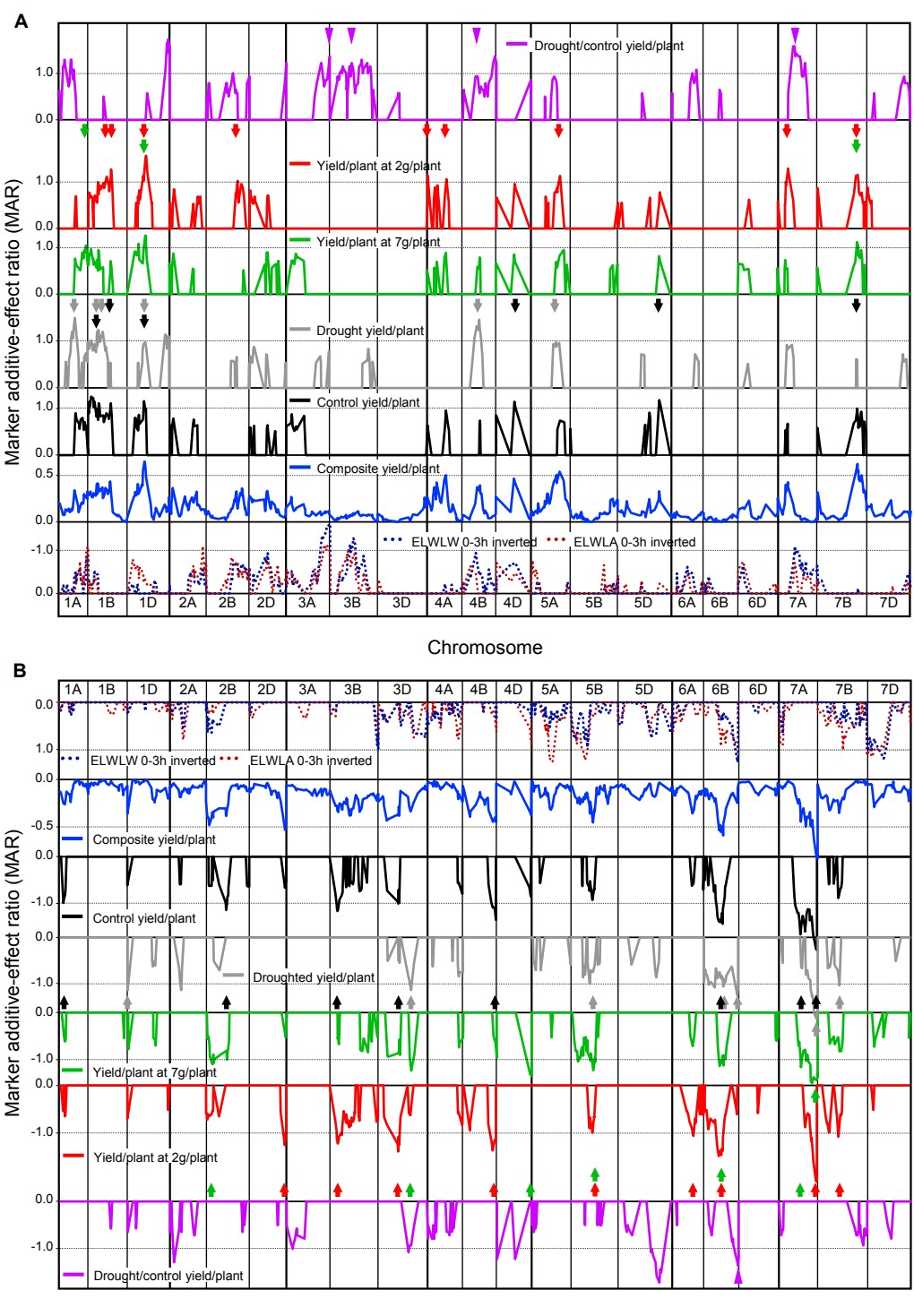

**Figure 4   Single marker analysis (SMA) of additive effects.** SMA expressed as marker additive ratios (MARs), for all 52 yield trials, control, droughted, yields at site yields of 7 t ha$^{-1}$ and 2 t ha$^{-1}$, droughted/-control yield ratio, as well as ELWLA$_{0-3\,h}$ and ELWLA$_{0-3\,h}$ (both inverted): (A) positive marker additive ratios (MARs) with increasing alleles from Chinese Spring, 

**Figure 4 (…continued)**
and (B) negative MARs with increasing alleles from SQ1. MARs for mean 52-trial yield, control and droughted yields were calculated using Method 1. For control yield, droughted yield, yields at site yields of $7\,\mathrm{t\,ha^{-1}}$, $2\,\mathrm{t\,ha^{-1}}$ and droughted/control yield ratio, to aid clarity, only MARs $\geq 0.5$ are shown. Coloured arrows indicate yield MAR peaks >1 coincident with peaks present for 52-trial MAR data. For 52-trial MAR data, 0.5 is equivalent to MARs greater than 1.0 in ca. 20% of trials and 0.35 is equivalent to MARs greater than 1.0 in ca. 10% of trials. Other details as described for Fig. 1.

**Table 7** Correlation coefficients for associations between marker additive effect ratio (MAR) maxima for ELWL and constitutive leaf traits and MARs at the same QTL markers for measures of yield.

| Trait | No of QTLs | All yields (52) | Controls (19) | Droughted (12) | Drought/control | 7 g/plant | 2 g/plant |
|---|---|---|---|---|---|---|---|
| ELWLW (0–3 h) | 25 | 0.042 | 0.305 | *−0.445*[*] | *−0.800*[****] | 0.231 | 0.057 |
| ELWLA (0–3 h) | 30 | *−0.399*[*] | 0.304 | *−0.574*[***] | *−0.365*[*] | *−0.680*[****] | 0.177 |
| ELWLW (3–6 h) | 24 | −0.001 | 0.267 | −0.402 | *−0.703*[****] | −0.205 | 0.254 |
| ELWLA (3–6 h) | 25 | −0.078 | −0.048 | −0.243 | −0.304 | −0.317 | −0.362 |
| Leaf length | 26 | **0.704**[****] | **0.502**[**] | **0.848**[****] | 0.342 | **0.771**[****] | **0.410**[*] |
| Leaf width | 24 | **0.639**[***] | **0.533**[**] | **0.416**[*] | −0.226 | **0.705**[****] | **0.505**[*] |
| Leaf area | 26 | **0.579**[**] | **0.422**[*] | **0.639**[****] | 0.227 | **0.665**[***] | 0.361 |
| Midrib thickness | 20 | 0.167 | 0.110 | 0.348 | 0.250 | 0.437 | 0.059 |

**Notes**
Significant positive correlations are shown in bold, and significant negative correlations are shown underlined.
[*], [**], [***], [****] indicate correlations significant at $P < 0.05$, $P < 0.01$, $P < 0.001$ and $P < 0.0001$, respectively, with 92 df.
Constitutive leaf trait QTLs listed in Table S1.

### Coincidence of QTLs between yield, ELWL and constitutive leaf traits

Peak MARs for four ELWL and four constitutive leaf traits (QTLs listed in Table S1) were used to test the likelihood of these traits determining one or other of the measures of yield. Thus, for each ELWL and leaf trait peak MAR (Table S1), the MAR was determined for each of the five measures of yield at either that marker or an adjacent marker if a yield MAR peak was within 10 cM and correlation coefficients were calculated for regressions of these yield peak MARs on peak MARs for the eight ELWL and leaf constitutive traits (Table 7).

As for phenotypic associations with yield, significant correlations of yield with ELWL traits were predominantly negative, and those with leaf constitutive traits predominantly positive. Significant correlations with leaf constitutive traits were more frequent than those with ELWL traits, and generally stronger. Using MARs, $\mathrm{ELWLA_{3-6\,h}}$ and leaf midrib thickness were not correlated with any measure of yield. Leaf 4 length was the trait most consistently highly correlated with measures of yield, except yield drought response (drought/control), illustrated for droughted yields in Fig. S4A.

In contrast, $\mathrm{ELWLW_{0-3\,h}}$ and $\mathrm{ELWLA_{0-3\,h}}$ were significantly negatively correlated with drought/control yield ratio, illustrated for $\mathrm{ELWLW_{0-3\,h}}$ in Fig. S4B and purple arrowheads in Fig. 4. Thus low water loss was associated with a relatively high yield under drought.

## DISCUSSION

### The physiological control of ELWL and its genetic variation

We selected leaf 4 from glasshouse-grown plants for our genetic analysis of ELWL, as the mapping population varies considerably in phenology (*Quarrie et al., 2005*), with days

to flag leaf emergence varying over two weeks (Table S5). Others have also studied leaf water loss in young plants (*Golestani Araghi & Assad, 1998*; *Rampino et al., 2006*), though *Clarke (1983)* found a genotype × environment interaction comparing glasshouse and field-grown plants sampled near anthesis.

ELWLW was usually highly negatively correlated with aspects of leaf size: leaf DW, leaf area, leaf length, as well as midrib thickness (Table 4), implying that a longer path length for water to reach the epidermis slowed rate of water loss from the leaf surface. Significant negative associations between rate of water loss and leaf area have been found by others in both sorghum (*Ali et al., 2011*) and wheat flag leaves (*Sayed & Bedawy, 2016*). Although variation in ELWLA could also reflect variation in leaf thickness, no significant relationship was found between ELWLA and leaf midrib thickness. Thus, distance from vascular bundles to the epidermis per se was unlikely to be a factor determining rate of water loss. The highly significant negative correlation between midrib thickness and $ELWLW_{0-3\,h}$ likely reflected the greater structural requirement of a thicker midrib as leaf length increased.

Stomatal number per unit area was not a factor in determining genotypic variation in ELWL amongst the CSDH lines (Table 4), though *Wang & Clarke (1993)* found a highly significant positive correlation between rate of water loss up to 2 h from excision and stomatal frequency amongst 12 hexaploid wheats. Furthermore, genotypic variation in the rate of water loss was unlikely to indicate genotypic variation in stomatal aperture as correlations for a given measure of ELWL between 0–3 h and 3–6 h (Table 4) were all significant, and stomata would be expected to have closed within a few minutes of leaf detachment as leaves lost turgor. Nevertheless, genotypic differences in non-stomatal water loss due to variation in cuticular thickness or composition, already reported for wheat (e.g., *Clarke & Richards, 1988*; *Jäger et al., 2014*; *Bi et al., 2016*; *Bi et al., 2017a*), could have contributed to the variation amongst CSDH lines in ELWL. It was not possible with the hand sections of Experiment III to assess cuticle thickness.

## The genetic control of ELWL and candidate genes

As additive effects using SMA varied up to 1.7-fold amongst experiments, additive ratios (Method 1) were used to compare between experiments and traits. Our QTL analyses (Table S1) demonstrated a broad genetic control of ELWL with QTLs distributed across several chromosomes, with increasing alleles from both parents, though few QTLs were stably expressed every year. MAR traces of ELWLW for different time intervals in 2007 (Fig. S6) implied the same genetic control of water loss for each time interval. Figure 1 confirms the extensive coincidence between 0–3 h and 3–6 h QTLs for both ELWLW and ELWLA. Therefore, it is probable that genetic variation in water loss was determined largely by non-stomatal characteristics. Although no cuticular traits were measured in our detached leaf experiments, visual assessment of CSDH line leaf waxiness at the tillering phase in the field in 2004 scored from 1 (very little visible wax) to 3 (thick greyish wax) showed QTLs coincident with those for $ELWLW_{0-3\,h}$ on 5BL and 5DL ($QELWLW_{0-3}.csdh\text{-}5B.2$ and $QELWLW_{0-3}.csdh\text{-}5D.1$) (Table S1).

*Yang et al. (2009)* and *Mei (2012)* have reported preliminary information on the genetic control of ELWL in wheat. *Yang et al. (2009)* identified QTLs on chromosomes 1D, 4A,

6B and 6D, with those on 6B (near locus *Xgwm193*) and 6D (near locus *Xbarc173*) being coincident with weak ELWL QTLs in our mapping population. *Mei (2012)* reported an additional QTL for ELWL on chromosome 2A where we also had a weak QTL effect.

Two likely classes of candidate gene for regulating leaf water loss would be those regulating water flow to the epidermis and those regulating its evaporation from the leaf surface through the cuticle. Aquaporins are water channel proteins belonging to the Major Intrinsic Protein superfamily of integral membrane proteins which specifically facilitate the passive flow of water molecules across cellular membranes (*Maurel, 1997*). *Forrest & Bhave (2010)* assigned several aquaporin genes to wheat chromosome bins. Plasma membrane aquaporin genes PIP1;1, PIP1;2, PIP2;2 and PIP2;1 were located in bins corresponding to ELWL QTLs on 2BS, 6AL and 7AS, respectively (Table S1). Tonoplast membrane aquaporins TIP1;2 and TIP2;1 were in bins corresponding to ELWL QTLs on 4BS and 6BL, respectively. An aquaporin gene listed in the GrainGenes wEST SQL bin-mapped markers database (http://wheat.pw.usda.gov/cgi-bin/westsql/map_locus.cgi; downloaded June 2006 as an Excel$^{MS}$ file and searched for "aquaporin"), BE403397, was located on bins C-2AL1-0.85, 6AL4-0.55-0.90 and 6BL5-0.40-1.00, each coincident with bins for ELWL QTLs (Table S1).

A recent publication by *Bi et al. (2017b)* reported the location of genes for several transcription factors regulating cuticle biosynthesis genes on the group 5 long arms, 6BL and 6DL. Two more genes with high sequence identity were found on 4A and 4D. QTLs for measures of ELWL were present on 5AL, 5BL, 5DL and 6BL. Only weak effects on ELWL were found on 4A and 4D. The well-characterised leaf waxiness genes *W1* and *Iw1* (*Wu et al., 2013*; *Hen-Avivi et al., 2016*) map distally on 2BS, where a QTL for ELWLA$_{3-6 h}$ was located (Table S1). Thus, some of the genes influencing ELWL may be associated with the regulation of water transport through aquaporins and genes for wax biosynthesis.

## ELWL as a trait for improving drought tolerance

Many authors have proposed excised-leaf water loss (or water retention), measured on the basis of either leaf water, fresh weight or dry weight, as a selection criterion to help improve drought tolerance (e.g., *Dedio, 1975*; *Clarke & McCaig, 1982b*; *Yang, Jana & Clarke, 1991*; *Dhanda & Sethi, 1998*; *David, 2010*). Indeed, significant positive relationships between excised-leaf water retention and yield have been found in studies on wheat genotypes under drought conditions (*Clarke et al., 1989*; *Petcu, 2005*; *Geravandi, Farshadfar & Kahrizi, 2011*), though not always (*Clarke et al., 1989*; *Dabiry et al., 2015*).

*Clarke & Townley-Smith (1986)* and *Clarke (1987)* demonstrated that selection for both high and low excised-leaf water retention in durum wheat crosses gave yield advantages for selections with low ELWL, but only under drought conditions. We therefore tested the efficacy of ELWL$_{0-3 h}$ as a selection criterion for yield in the CSDH population and compared this with leaf 4 length, a much simpler trait to measure and one very similar to ELWL in its phenotypic correlations with yield under both control and droughted conditions (Tables 5A, 5B). Although we demonstrated a yield advantage for the 10 CSDH lines with both the lowest ELWL$_{0-3 h}$ and longest leaves compared with the 10 lines at the opposite end of the trait rankings, as a breeding criterion, the advantage of selecting for

low $ELWL_{0-3\,h}$ was much less apparent (Table 6), with an overall yield advantage of only around 3% compared with the remaining lines in the population, with no clear additional benefit under droughted conditions. Therefore, as a selection criterion, $ELWL_{0-3\,h}$ would probably be no more effective than leaf 4 length in improving wheat yields.

## CONCLUSIONS

To conclude, our genetic analysis of ELWL showed many regions of the wheat genome to contribute to variation in water loss, with few dominant and stably expressed QTLs. Only 13% of QTLs for ELWL traits reached significance in every year (Table S1). Nevertheless, $ELWLW_{0-3\,h}$ and $ELWLA_{0-3\,h}$ were significantly negatively correlated with grain yield, but irrespective of water status. As a yield selection criterion, $ELWLW_{0-3\,h}$ and leaf length were equally effective.

### Funding

The study was partly financed by the Ministry of Science and Higher Education, Poland, (No 479/N-COST-2009/0 and 480/N-COST/2009/0), statutory investigations in IPP PAS, Kraków, Poland. The manuscript was prepared within a bilateral cooperation project between Polish and Serbian Academies of Science and Faculty of Agriculture, Belgrade University, 2016–2019. There was no additional external funding received for this study. The funders had no role in study design, data collection and analysis, decision to publish, or preparation of the manuscript.

### Grant Disclosures

The following grant information was disclosed by the authors:
Ministry of Science and Higher Education, Poland: 479/N-COST-2009/0, 480/N-COST/2009/0.
Polish and Serbian Academies of Science and Faculty of Agriculture, Belgrade University.

### Competing Interests

The authors declare there are no competing interests.

### Author Contributions

- Ilona Mieczysława Czyczyło-Mysza and Stephen Alexander Quarrie conceived and designed the experiments, performed the experiments, analyzed the data, contributed reagents/materials/analysis tools, prepared figures and/or tables, authored or reviewed drafts of the paper, approved the final draft.
- Izabela Marcińska conceived and designed the experiments, performed the experiments, contributed reagents/materials/analysis tools, approved the final draft.
- Edyta Skrzypek, Kinga Dziurka and Dejan Dodig performed the experiments, approved the final draft.
- Jan Bocianowski analyzed the data, approved the final draft.

- Dragana Rančić and Radenko Radošević performed the experiments, contributed reagents/materials/analysis tools, approved the final draft.
- Sofija Pekić-Quarrie conceived and designed the experiments, approved the final draft.

## Data Availability

The raw data are provided in the Supplemental Files.

## Supplemental Information

Supplemental information for this article can be found online at http://dx.doi.org/10.7717/peerj.5063#supplemental-information.

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
