# Peer review of "Genetic analysis of water loss of excised leaves associated with drought tolerance in wheat"

_PeerJ, doi:10.7717/peerj.5063_

## Round 0.1 · original submission · Minor Revisions

There are very minor remarks from the reviewers, please fix them.

·

Basic reporting

Basic reporting is fine

Experimental design

Experimental design is fine

Validity of the findings

The finding is valuable and fine

Additional comments

Minor corrections/suggestions:

(1) The title of this manuscript has to be modified and shorten. This is not suitable to use two phrases in one title: ‘water loss of excised wheat leaves’ and ‘excised-leaf water loss’. Author can choose and keep either first or second part of the title only. Alternatively, it is possible to merge two parts of the title together with the elimination of repeated phrases.

(2) L54, L56, L59, L527, L530, and in entire manuscript. It would be better to use a term ‘drought tolerance’ rather than ‘drought resistance’ because ‘resistance’ is more suitable for ‘disease resistance’.

(3) L186. Please specify what types of 702 molecular markers were used for genetic mapping and QTL analysis? In the indicated Reference, DArT markers were used but earlier SSR markers were employed. Please indicate, if only these or other molecular markers were present in the used genetic map.

(4) L234, L241, L426 and in other parts. Please use a special mark ‘cross’ indicating a hybridization and hybrid analysis rather than letter ‘x’. The correct marks were used on Lines 238 and 289.

(5) L387. Please indicate, what does ‘constitutive’ mean in the context of leaf traits? This term can confuse Readers for ‘constitute gene expression’. Therefore, it would be better to replace ‘constitute’ to another term with similar meaning.

(6) L428-434 and in other parts of the manuscript. Please explain and modify accordingly to make clear phrases ‘increasing allele’ and ‘alleles increasing’. Is this increased frequency of a certain allele in analysed progeny? Increased probability or something else? The meaning of these phrases must be very clear for all Readers.

·

Basic reporting

- The manuscript (MS) by Czyczyło-Mysza et al. is written in very good English, both fluent and technically correct.

- It contains a sufficient Introduction for the reader to enter the research topic; subsequent sections are also well structured, enabling the reader to follow the experimental and analytical work done.

- Figures and Tables are generally well done and correct. However, numbering has to be checked. In fact, no Fig. 4b was included in the Reviewing PDF, but it is quoted on page 14 of the MS (under “Genetic analysis of yield per plant” heading). As only Figure 4a seems to exist, it should be named Figure 4. In the Reviewing PDF, what is called Figure 5 legend (probably meant as legend to Fig. 4) seems incomplete, and some formatting problems were also found in other Figures (e.g. 1 and 2). As no separate Figure legends file or page was found to check this, the authours/journal should verify the correctness/completeness of the legend content.
Also, Table S1 is quoted in several places in the MS, starting from page 15 under ”Coincidence of QTLs between yield, ELWL and constitutive leaf traits”, but there is no such Table in the Supplementary materials…
A slightly different formatting is suggested for Figure 5 (attached PDF, in place of annotated MS).

Experimental design

The investigation has been conducted rigorously, with a comprehensive description of methods

Validity of the findings

A very large amount of data is provided (including the raw data), and results from statistical analyses are sound. Hence the conclusions are fully justified.

Additional comments

The paper provides an interesting and useful piece of information, which contributes to clarify the relative validity of selection criteria to be applied under normal and droughted wheat growing conditions.
It definitely deserves publication, with virtually no modification, except for the very minor ones concerning Figures and Tables, as indicated in the "Basic reporting" section.

---

## Round 0.2 · accepted · Accept

Thank you for the updated manuscript. The reviewer had no more remarks. I believe the paper should be accepted in current form.

# ·

Basic reporting

Fine

Experimental design

Fine

Validity of the findings

Fine

Additional comments

All corrections are fine